# The yeast mitochondrial pyruvate carrier is a hetero-dimer in its functional state

Sotiria Tavoulari[1,*] (ID), Chancievan Thangaratnarajah[1,†], Vasiliki Mavridou[1], Michael E Harbour[1], Jean-Claude Martinou[2] & Edmund RS Kunji[1,**] (ID)

## Abstract

The mitochondrial pyruvate carrier (MPC) is critical for cellular homeostasis, as it is required in central metabolism for transporting pyruvate from the cytosol into the mitochondrial matrix. MPC has been implicated in many diseases and is being investigated as a drug target. A few years ago, small membrane proteins, called MPC1 and MPC2 in mammals and Mpc1, Mpc2 and Mpc3 in yeast, were proposed to form large protein complexes responsible for this function. However, the MPC complexes have never been isolated and their composition, oligomeric state and functional properties have not been defined. Here, we identify the functional unit of MPC from *Saccharomyces cerevisiae*. In contrast to earlier hypotheses, we demonstrate that MPC is a hetero-dimer, not a multimeric complex. When not engaged in hetero-dimers, the yeast Mpc proteins can also form homo-dimers that are, however, inactive. We show that the earlier described substrate transport properties and inhibitor profiles are embodied by the hetero-dimer. This work provides a foundation for elucidating the structure of the functional complex and the mechanism of substrate transport and inhibition.

**Keywords** mitochondria; oligomeric state; protein complex; pyruvate; transport proteins
**Subject Categories** Membrane & Intracellular Transport
The EMBO Journal (2019) 38: e100785

## Introduction

In recent years, there is an increasing understanding and appreciation that mitochondrial metabolism is involved in major human diseases, such as cancer, neurodegeneration, cardiovascular diseases, metabolic disorders, obesity and diabetes. A key player for the metabolic fate of the cell is the mitochondrial pyruvate carrier (MPC), a protein responsible for the uptake of pyruvate from the cytosol into the mitochondrial matrix (Vanderperre *et al*, 2015), where it enters the tricarboxylic acid cycle and other biosynthetic pathways. The existence of a membrane protein responsible for pyruvate transport across the mitochondrial inner membrane had been supported by early work on isolated mitochondria, where pyruvate transport had been shown to be saturating and pH-dependent (Papa *et al*, 1971; Papa & Paradies, 1974). In addition, small-molecule inhibitors had been identified in support of this notion (Halestrap & Denton, 1974; Halestrap, 1975, 1976), but the molecular identity of the protein remained unknown for four decades.

Major progress was made in 2012, when mitochondrial pyruvate transport activity was discovered to be associated with two small homologous proteins, MPC1 and MPC2 (Bricker *et al*, 2012; Herzig *et al*, 2012). In mammals and *Drosophila*, the expression of both MPC proteins is necessary for pyruvate transport (Bricker *et al*, 2012; Herzig *et al*, 2012). In yeast *Saccharomyces cerevisiae*, three proteins Mpc1, Mpc2 and Mpc3 are expressed in a carbon source-dependent pattern, forming an Mpc1/Mpc2 complex under fermentative conditions and an Mpc1/Mpc3 under respiratory conditions, called MPC$_{FERM}$ and MPC$_{OX}$ complexes, respectively (Bender *et al*, 2015; Compan *et al*, 2015).

As the mitochondrial pyruvate carrier is central for cellular homeostasis, the identification of the MPC proteins intensified efforts to understand their role in cancer (Schell *et al*, 2014, 2017; Yang *et al*, 2014; Zhong *et al*, 2015; Li *et al*, 2016, 2017; Corbet *et al*, 2018; Ohashi *et al*, 2018; Bader *et al*, 2019), diabetes (Colca *et al*, 2013; Divakaruni *et al*, 2013; Vigueira *et al*, 2014; Gray *et al*, 2015; McCommis *et al*, 2015, 2016; Vadvalkar *et al*, 2017) and neurodegeneration (Ghosh *et al*, 2016; Divakaruni *et al*, 2017; Quansah *et al*, 2018). Moreover, pathogenic mutations in the *mpc1* gene were found in rare but severe metabolic syndromes, further enhancing the clinical relevance of this transporter (Bricker *et al*, 2012). Additionally, an increasing number of small-molecule drugs, previously known to have other targets, have now been proposed to inhibit MPC activity (Colca *et al*, 2013; Divakaruni *et al*, 2013; Du *et al*, 2013; Ghosh *et al*, 2016; Nancolas *et al*, 2016; Nath *et al*, 2016; Chen *et al*, 2018; Corbet *et al*, 2018). MPC is a newly identified target for the first-generation insulin sensitisers, called thiazolidinediones (TZDs; Colca *et al*, 2013; Divakaruni *et al*, 2013),

1 Medical Research Council Mitochondrial Biology Unit, University of Cambridge, Cambridge, UK
2 Department of Cell Biology, University of Geneva, Genève 4, Switzerland
  *Corresponding author. Tel: +441223252850; Fax: +441223252875; E-mail: st632@mrc-mbu.cam.ac.uk
  **Corresponding author. Tel: +441223252850; Fax: +441223252875; E-mail: ek@mrc-mbu.cam.ac.uk
  †Present address: Groningen Biomolecular Sciences and Biotechnology Institute, Membrane Enzymology, University of Groningen, Groningen, The Netherlands

originally known to exert their action on the peroxisome proliferator-activated receptor gamma (PPARγ; Soccio *et al*, 2014; Nanjan *et al*, 2018). More recently, a new generation TZD, bypassing PPARγ (Colca *et al*, 2014) and currently in clinical trials for the treatment of Parkinson's disease, was proposed to exert its action by inhibiting MPC (Ghosh *et al*, 2016). Despite these major advances, a direct experimental system to measure the interactions of small-molecule drugs with MPC and to study the mechanism of inhibition is not available.

Six years after the primary identification of the MPC proteins (Bricker *et al*, 2012; Herzig *et al*, 2012), there has been no report of a successful purification and functional reconstitution of an MPC hetero-complex. Consequently, the composition of the MPC complexes, the oligomeric state and the protomer stoichiometry remain controversial (Bricker *et al*, 2012; Bender *et al*, 2015; Nagampalli *et al*, 2018) and their involvement in pyruvate transport has been questioned (Halestrap, 2012). The yeast MPC hetero-complexes migrate at 150 kDa (Bricker *et al*, 2012; Bender *et al*, 2015) or at even higher molecular weights (Bender *et al*, 2015) in blue native gel electrophoresis, leading to proposals that the complexes are multimeric and might even include additional, yet unidentified proteins (Bricker *et al*, 2012; Halestrap, 2012). The application of chemical cross-linking on MPC proteins has produced bands corresponding to monomers, dimers and higher oligomers (Bender *et al*, 2015; Nagampalli *et al*, 2018). In the only published attempt to purify an MPC hetero-complex, it was only possible to purify individual MPC protomers (Nagampalli *et al*, 2018). It has also been proposed that the individual human MPC2 protein can form high-order multi-species capable of transporting pyruvate (Nagampalli *et al*, 2018), raising more questions regarding the functional unit of the mitochondrial pyruvate carrier.

Here, we report the first successful purification and characterisation of MPC hetero-complexes from yeast, providing a model system for future structural and mechanistic studies. We demonstrate that the natural state of yeast MPC is a hetero-dimer capable of transporting pyruvate. In the absence of other protomers, MPC proteins can form homo-dimers, but they do not transport pyruvate.

# Results

### Expression and purification of the yeast Mpc proteins

Although it has been established that the MPC proteins constitute the mitochondrial pyruvate carrier (Bricker *et al*, 2012; Herzig *et al*, 2012), there are still many outstanding questions regarding the composition and the oligomeric state of the functional complex. The most straightforward way to settle these issues is through the purification and reconstitution of the individual components and putative complexes to determine whether they have pyruvate transport activity.

Here, we have used the Mpc proteins from *S. cerevisiae* (Fig EV1) as a model system to study the composition and functional properties of the mitochondrial pyruvate carrier. We decided to concentrate on the Mpc1/Mpc3 complex, as it is the principle pyruvate carrier in oxidative phosphorylation (MPC$_{OX}$; Bender *et al*, 2015). It is also most related to MPCs of mammals and other organisms, whereas the Mpc1/Mpc2 complex (MPC$_{FERM}$), expressed under fermentative conditions, only exists in some fungi. The

principal approach was to purify the Mpc1/Mpc3 hetero-complex by tagging one of the two protomers with a Factor Xa cleavage site and an eight-histidine tag on the C-terminus (Fig 1A). To achieve co-expression of the protein pair, we used an inducible bidirectional vector (Miller *et al*, 1998) for expression in mitochondria of the *mpc* triple deletion strain SHY15 (Herzig *et al*, 2012). We also expressed the C-terminally tagged Mpc1 and Mpc3 proteins individually in yeast mitochondria.

All proteins expressed well, both in the presence and in the absence of their proposed complex partner (Fig 1B). It is notable though that in expression trials of Mpc1 alone an additional prominent band appeared, approximately corresponding to the molecular weight of an SDS-resistant dimer, as detected by immunoblotting of crude yeast mitochondrial preparations (Fig 1B, left panel). However, this band was not detected in mitochondria expressing the untagged Mpc1 protein (Fig 1B, left panel). For these reasons, we chose to co-express an unmodified Mpc1 together with a tagged Mpc3 for hetero-complex formation.

With this strategy, the purification of the Mpc1/Mpc3 hetero-complex was successful and both proteins were present in a 1:1 ratio (Fig 1C). The highest protein yield (~1 mg protein per g of mitochondria) was achieved with the detergent lauryl maltose neopentyl glycol (LMNG), supplemented with tetraoleoyl cardiolipin. However, the stoichiometric Mpc1/Mpc3 complex could also be purified in Triton X-100, decyl maltose neopentyl glycol (DMNG) or *n*-dodecyl β-D-maltoside (DDM), each supplemented with tetraoleoyl cardiolipin, albeit with lower purification yields (Fig EV2). In LMNG and tetraoleoyl cardiolipin, the Mpc1/Mpc3 complex eluted as a single peak by size-exclusion chromatography, indicating that the complex was monodisperse (Fig EV3). Moreover, analysis of the peak fractions showed that the protomers remained associated during purification, consistent with a stable complex (Fig EV3A, inset). When we purified histidine-tagged Mpc1 or Mpc3 on their own in LMNG and tetraoleoyl cardiolipin (Fig 1C), the yields were at least three times lower, despite similar expression levels in mitochondria in the presence or absence of their partner (Fig 1B), indicating stability issues. After purification and histidine-tag cleavage, Mpc1 contained again an SDS-resistant dimer (Fig 1C), as detected by peptide mass fingerprinting (Table EV1).

Next, we used thermostability analysis to evaluate folding and stability of the hetero-complex and its components. For this purpose, we monitored the unfolding of protein populations in a temperature ramp in the presence of 7-diethylamino-3-(4-maleimidophenyl)-4-methylcoumarin (CPM), which reacts with cysteine residues becoming exposed due to denaturation (Fig 1D). Mpc1 and Mpc3 both have a single cysteine residue (Fig EV1). The Mpc1/Mpc3 hetero-complex and the individual Mpc3 protein had very similar unfolding profiles and melting temperatures (52.0 and 51.7°C, respectively), clearly showing that they are folded and stable in detergent solution. The Mpc1 protein on its own, however, did not display a thermal denaturation profile, and high fluorescence was detected throughout the temperature ramp. This result means that either Mpc1 is unfolded or its single cysteine is already exposed. To discriminate between these two possibilities, we also performed thermostability analysis by nano differential scanning fluorimetry (nanoDSF; Fig 1E), which relies on changes in the environment of endogenous tyrosine and tryptophan residues. Again, the hetero-complex and Mpc3 had similar apparent melting

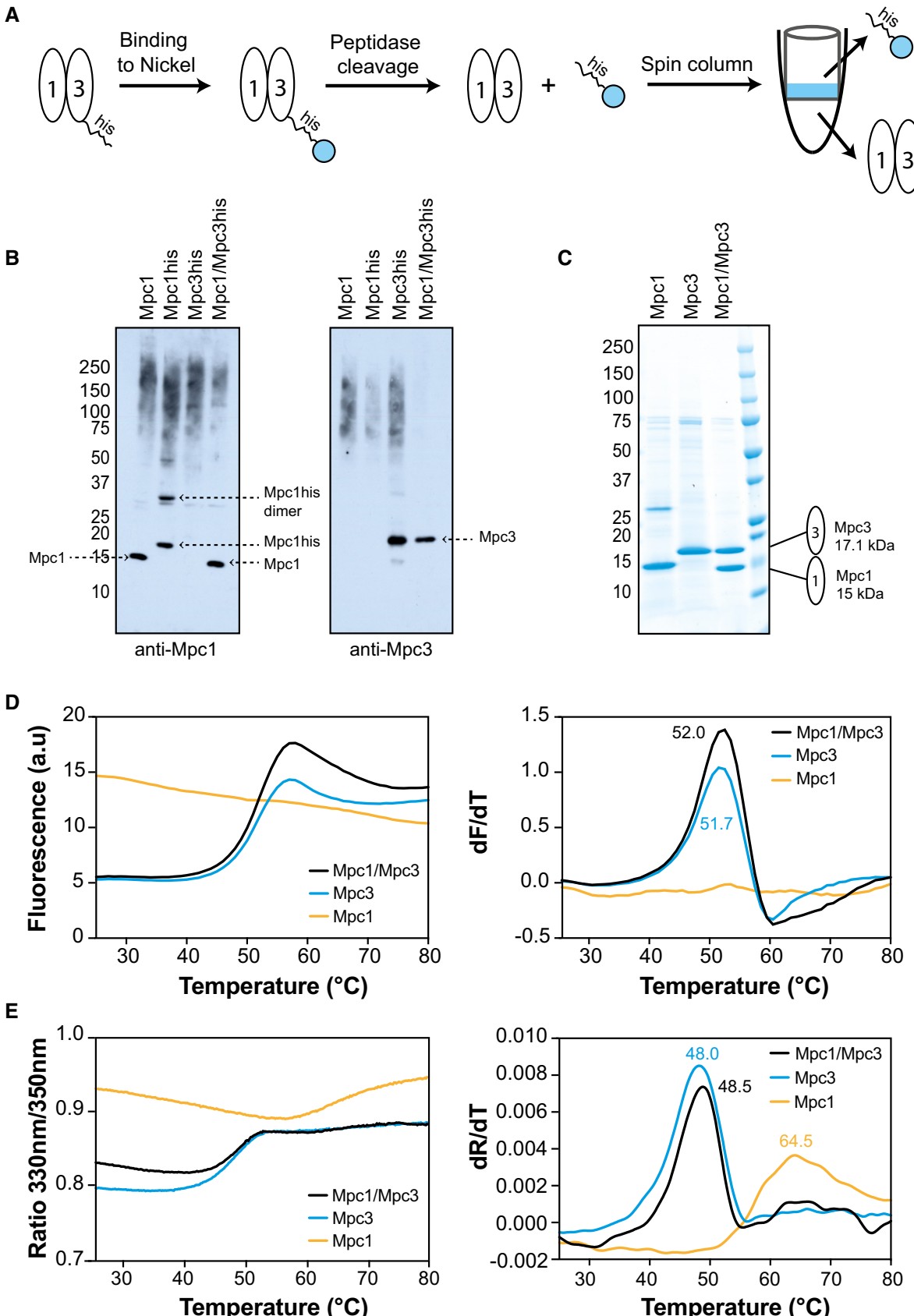

**Figure 1.**

**Figure 1.  Purification and stability analysis of Mpc proteins.**

A   Strategy for purification of the Mpc1/Mpc3 hetero-complex by nickel-affinity chromatography.
B   Expression of Mpc proteins in mitochondria assessed by SDS–PAGE and immunoblot analysis of crude mitochondrial preparations. The individual untagged Mpc1 (Mpc1), histidine-tagged Mpc1 (Mpc1his), histidine-tagged Mpc3 (Mpc3his) or the Mpc1/Mpc3 hetero-complex (Mpc1/Mpc3his) were detected with antibodies raised against Mpc1 (left panel) or Mpc3 (right panel) and are shown with dashed arrows.
C   Five micrograms of each affinity-purified Mpc protein were analysed by SDS–PAGE, and the bands were visualised by Coomassie Blue stain. Peptide mass finger printing was used to identify the major protein bands (Table EV1).
D   The stability of the purified proteins was assessed via thermal denaturation by fluorescent CPM-adduct formation. The thermal denaturation profiles (left) were used to calculate the first derivative (right), which provides the apparent melting temperature, indicated with the same colour coding.
E   The stability of the same samples, as in panel (D), was assessed by NanoDSF. The changes in the 330 nm/350 nm ratio with temperature (left) were used to calculate the first derivative (right). Colour coding is as in panel (D).

temperatures, 48.5 and 48.0°C, respectively, similar to those obtained with CPM. However, Mpc1 alone showed a peak at 64.5°C, which might correspond to the SDS-resistant dimer, possibly being an aggregation artefact.

The ability of Mpc proteins to form hetero-complexes was also demonstrated when we expressed and purified the Mpc1/Mpc2 (MPC$_{FERM}$) hetero-complex (Bender *et al*, 2015) using a histidine tag on Mpc2 (Fig EV4A and B). We also compared the Mpc1/Mpc2 hetero-complex to the histidine-tagged Mpc2, expressed and purified alone (Fig EV4). Although Mpc2 can be expressed in mitochondria equally well on its own or together with Mpc1 (Fig EV4A), it could not be purified alone in sufficient quantities and could only be detected by peptide mass fingerprinting (Fig EV4B and Table EV1). Interestingly though, when they were expressed together, Mpc2 with Mpc1 successfully formed a hetero-complex, which was purified and had an apparent melting temperature of 42°C (Fig EV4C).

This is the first purification of MPC hetero-complexes, demonstrating that different Mpc protomers can form stable interactions. Of the two hetero-complexes, the principal complex Mpc1/Mpc3 was purified at higher yields, was 10°C more thermostable and was, therefore, selected for characterisation of its oligomeric state.

**The Mpc1/Mpc3 hetero-complex is a dimer**

We showed that the protomers are present in a 1:1 molar ratio in the hetero-complexes (Fig 1C), but this does not resolve the issue of the overall mass of the complex. Previous work on the yeast Mpc proposed that the protein is a multimeric complex of 150 kDa, based on its electrophoretic mobility by blue native gel electrophoresis (Bricker *et al*, 2012) and this notion has been largely accepted in the literature. However, the electrophoretic mobility on blue native gels depends on the associated detergent and lipids present during protein extraction, as both the detergent-lipid micelle (DL) and the protein bind Coomassie stain, leading to anomalous migration (Crichton *et al*, 2013). Therefore, blue native gel electrophoresis is not an appropriate technique for sizing of small membrane proteins. Another approach that has been used involves chemical cross-linking with 7 and 15 Å long cross-linkers. The cross-linked MPC proteins appeared on SDS–PAGE in multiple bands corresponding to monomers, dimers and different higher oligomers (Bender *et al*, 2015; Nagampalli *et al*, 2018), but it cannot be excluded that the detected states are the result of non-specific cross-linking events.

Here, we determined the molecular mass, oligomeric state and subunit stoichiometry of the affinity-purified Mpc1/Mpc3 hetero-complex by using size-exclusion chromatography linked to multi-angle laser light scattering (SEC-MALLS; Fig 2A and B). This

technique can determine the mass of the protein–detergent–lipid complex (PDL) and of the protein itself (Slotboom *et al*, 2008; ter Beek *et al*, 2011). The analysis showed that on average the mass of the Mpc1/Mpc3 hetero-complex is contributing 30.8 ± 1.4 kDa to a 163.3 ± 5.1 kDa protein–detergent–lipid complex (Table EV2 and Fig 2B). The protein mass corresponded well to the sum of the theoretical masses of Mpc1 (15 kDa) and Mpc3 (17.1 kDa), which demonstrates that the complex is a hetero-dimer. Similar results were obtained whether the protein was purified by a single affinity chromatography step or by an additional size-exclusion chromatography step (Table EV2).

To exclude other possible stoichiometries, we performed an internal consistency analysis (Wen *et al*, 1996), where theoretical molecular masses for the complex, corresponding to different possible subunit stoichiometries, were calculated and compared with the experimentally determined molecular mass. The analysis was only consistent with a Mpc1/Mpc3 hetero-dimeric complex, showing the smallest difference between the experimentally determined and theoretical molecular masses (Table 1). Additionally, since Mpc1 and Mpc3 co-purify in equimolar amounts and remain associated throughout the SEC-MALLS step (Fig 2B, inset), the possibility that the calculated mass corresponds to separate homo-dimers is eliminated.

Since Mpc3 can be purified on its own (Figs 1C and EV3) and is stable in solution, it was also analysed by SEC-MALLS (Fig 2C and D). The average molecular mass for the protein–detergent–lipid complex was 182.9 ± 7.9 kDa with the protein contributing 34.8 ± 0.7 kDa (Table EV2 and Fig 2D), which is approximately twice the theoretical mass of Mpc3 (17.1 kDa). Thus, Mpc3 forms homo-dimers in the absence of Mpc1.

**The Mpc1/Mpc3 hetero-dimer is the active mitochondrial pyruvate carrier**

To investigate whether the Mpc1/Mpc3 hetero-dimer is capable of pyruvate transport, we established a reconstitution and transport protocol. We prepared proteoliposomes of the purified hetero-dimer loaded with 5 mM unlabelled pyruvate in buffer at pH 8.0 (Fig 3) or 7.4 (Fig EV5), and we initiated pyruvate homo-exchange by addition of radiolabelled pyruvate (50 μM) on the outside. To evaluate the pH dependence of transport, we performed our assays with external buffer at different pH units (Figs 3 and EV5).

In proteoliposomes loaded with internal buffer at pH of 7.4 (Fig EV5), we observed that diffusion, as determined by [$^{14}$C]-pyruvate accumulation in empty liposomes, was pH-dependent and highest in acidic pH of 5.4, when pyruvate accumulation into empty liposomes reached levels even higher than into proteoliposomes.

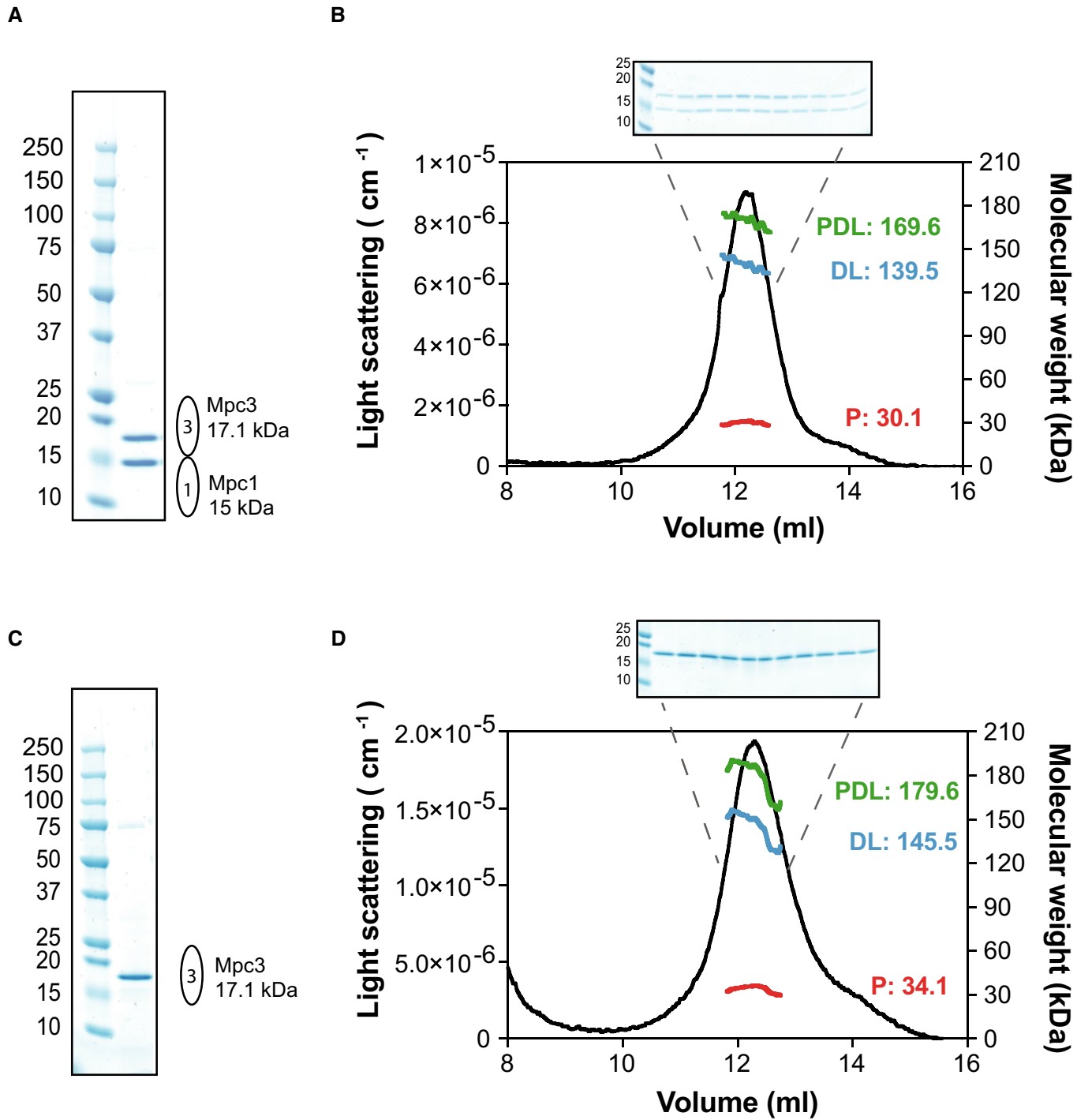

**Figure 2. The Mpc proteins form dimeric complexes.**

A   Nickel-affinity-purified Mpc1/Mpc3 hetero-complex used for SEC-MALLS analysis, showing a 1:1 stoichiometry of the protomers.
B   SEC-MALLS analysis of the hetero-complex. The light scattering trace for Mpc1/Mpc3 is shown as a black line. The masses of the protein–detergent–lipid complex (PDL, green), the detergent–lipid micelle (DL, blue) and the protein (P, red) are indicated. Protein fractions across the peak were assessed by SDS–PAGE and visualised by Coomassie Blue staining (B, inset).
C   Nickel-affinity-purified Mpc3 protein used for SEC-MALLS analysis.
D   SEC-MALLS analysis of Mpc3, colour designation as in (B). The protein fractions across the peak (D, inset) were assessed as in (B).

Data information: Data in (B and D) represent a characteristic experiment repeated independently five times for Mpc1/Mpc3 and three times for Mpc3. All biological repeats are summarised in Table EV2.

Table 1. Stoichiometry analysis of the Mpc1/Mpc3 complex.

| Mpc1 | Mpc3 | Absorbance 0.1% (=1 g/l) | *In silico* calculated $M_w$ (kDa) | $M_w$ from SEC-MALLS (kDa) | Difference (kDa) |
|---|---|---|---|---|---|
| Monomer | | | | | |
| 1 | 0 | 1.628 | 15 | 32.92 | 17.92 |
| 0 | 1 | 1.868 | 17.12 | 28.69 | 11.57 |
| Homo-dimer | | | | | |
| 2 | 0 | 1.633 | 29.97 | 32.9 | 2.93 |
| 0 | 2 | 1.869 | 34.21 | 28.68 | −5.53 |
| Hetero-dimer | | | | | |
| 1 | 1 | 1.757 | 32.1 | 30.83 | −1.27 |
| Homo-trimer | | | | | |
| 3 | 0 | 1.629 | 44.95 | 32.9 | −12.05 |
| 0 | 3 | 1.869 | 51.31 | 29.6 | −21.71 |
| Hetero-trimer | | | | | |
| 2 | 1 | 1.716 | 47.07 | 31.21 | −15.86 |
| 1 | 2 | 1.796 | 49.19 | 30.12 | −19.07 |

This result is consistent with previous studies suggesting that pyruvate can cross the mitochondrial membrane via "absorption", depending on its protonation state (Klingenberg, 1970; Zahlten *et al*, 1972; Bakker & van Dam, 1974). Therefore, the selection of internal and external pH units is critical. Our finalised pyruvate transport protocol (Fig 3) was based on achieving the maximal ΔpH under conditions where the diffusion is minimal.

When we selected an internal pH of 8.0 and external pH of 6.4, yielding a ΔpH of 1.6, high pyruvate homo-exchange activity was observed (Fig 3A) with an initial uptake rate of $1.6 \pm 0.2$ μmol/min/mg protein, demonstrating that the purified Mpc1/Mpc3 hetero-dimer is capable of transporting pyruvate. This homo-exchange reaction was completed within seconds when performed at room temperature. Under these conditions, the maximum transport rate ($V_{max}$) was $8 \pm 3$ μmol/min/mg protein and the apparent $K_M$ of transport for pyruvate was $342 \pm 58$ μM (Fig 3B). We were also able to measure homo-exchange activity at a ΔpH of 0.8 (internal pH of 8.0, external pH of 7.2), which is physiologically relevant to mitochondria, albeit with lower signal (Fig 3C). The activity was abolished in the absence of a ΔpH (Fig 3D), showing that transport is pH-dependent, as previously proposed for the carrier in mitochondria (Papa *et al*, 1971; Papa & Paradies, 1974). Importantly, the Mpc1/Mpc2 hetero-complex was also active for ΔpH-dependent pyruvate transport (Fig EV4D).

It is expected that transport by a functional MPC should be inhibited by UK5099, a well-established inhibitor of pyruvate transport in rat, mouse or human mitochondria (Halestrap, 1975). We tested whether UK5099 can also inhibit pyruvate homo-exchange by the yeast Mpc1/Mpc3 hetero-dimer. Indeed, in the presence of UK5099 the initial rates of transport were inhibited, with the $IC_{50}$ being at the low micromolar range ($9 \pm 7$ μM from three biological repeats and Fig 3E). This is a greater value than reported for mammalian MPC proteins, where the $IC_{50}$ for UK5099 is in the nanomolar range, probably reflecting differences in inhibitor binding between yeast and mammalian complexes. We also tested Zaprinast, a phosphodiesterase inhibitor, which was reported to block pyruvate oxidation and pyruvate influx in mouse mitochondria (Du *et al*, 2013), and 7ACC2, a carboxycoumarin inhibitor of the monocarboxylate transporter 1 (MCT1), also proposed to inhibit MPC. The inhibition of pyruvate exchange was similar for Zaprinast and 7ACC2 with an average $IC_{50}$ of $18 \pm 8$ and $27 \pm 13$ μM, respectively. Lonidamine, an anti-tumour agent proposed to inhibit the monocarboxylate transporters (MCTs) as well as MPC, inhibited the yeast hetero-complex but with a higher $IC_{50}$ of $118 \pm 24$ μM. Finally, we tested two thiazolidinediones (Fig 3F). Unlike the other tested compounds, pioglitazone had no effect on pyruvate transport by Mpc1/Mpc3. Rosiglitazone inhibited pyruvate transport to $52 \pm 7\%$ but at a high concentration (500 μM) suggesting that some TZDs might be inhibiting the yeast complex, albeit with very low affinity. Taken together, our work shows that the reconstituted Mpc1/Mpc3 hetero-dimer displays the expected characteristics of the mitochondrial pyruvate carrier, providing the first direct experimental evidence of the functional unit.

**Individual Mpc proteins form homo-dimers but they are not functional**

In the absence of other Mpc proteins, Mpc3 formed a stable homo-dimer (Fig 2C and D). The human MPC2 protein, which is related to Mpc3, was recently purified and reconstituted into liposomes and was proposed to have pyruvate transport activity (Nagampalli *et al*, 2018). To clarify whether the individual yeast Mpc proteins are capable of pyruvate transport, under conditions where their partner is not expressed, we purified and reconstituted Mpc1 and Mpc3 separately in liposomes and tested whether they could transport pyruvate. In parallel, the Mpc1/Mpc3 hetero-complex was used as control. While the hetero-complex mediated robust pyruvate homo-exchange at a ΔpH of 1.6, there was no measurable activity for Mpc3 or Mpc1 alone (Fig 4A and B). It is unlikely that this result is due to differences in reconstitution. First, the purified Mpc3 homo-dimer is as stable in detergent as the functional hetero-dimer (Fig 1D and E), and second, a similar amount of Mpc3 was reconstituted into liposomes as the Mpc1/Mpc3 complex.

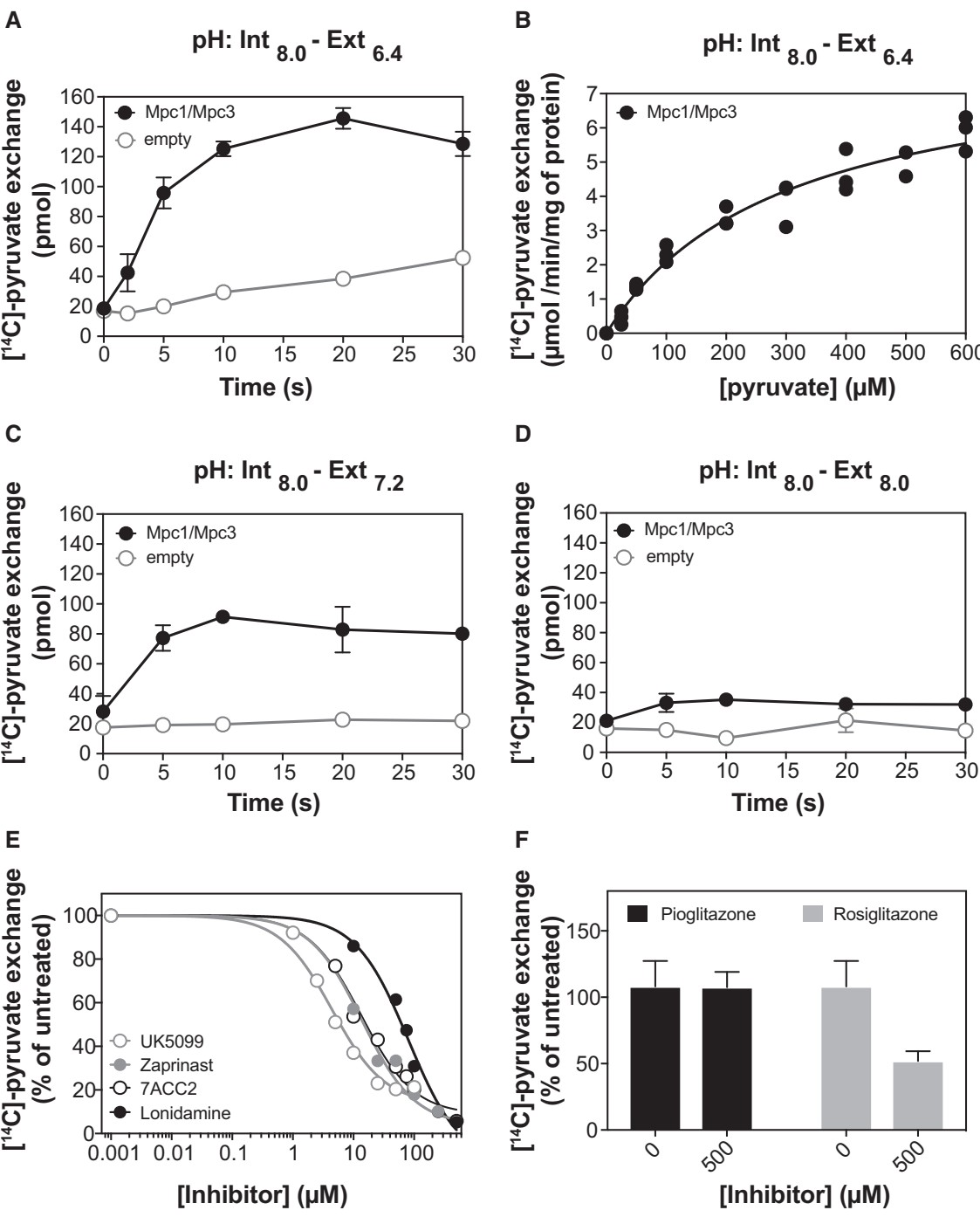

**Figure 3. The Mpc1/Mpc3 hetero-complex transports pyruvate.**

A   Time course of pyruvate homo-exchange by the Mpc1/Mpc3 hetero-complex in liposomes ($n = 8$) in comparison with empty liposomes ($n = 6$) at a $\Delta$pH of 1.6.

B   Kinetic analysis of pyruvate homo-exchange at $\Delta$pH of 1.6 ($n = 3$). The hetero-complex was assayed for initial rates of uptake in the concentration range of 25–600 $\mu$M. The calculated $K_M$ was 299 $\mu$M in this experiment and 318 and 409 $\mu$M in two additional biological repeats.

C   Time course of pyruvate homo-exchange by the Mpc1/Mpc3 hetero-complex in physiological pH ($n = 4$) compared to empty liposomes ($n = 4$).

D   In the absence of a $\Delta$pH, the time course of pyruvate homo-exchange was similar for the Mpc1/Mpc3 proteoliposomes and the empty liposomes ($n = 4$).

E   Inhibition of [$^{14}$C]-pyruvate homo-exchange by UK5099 (1–100 $\mu$M), Zaprinast (1–1,000 $\mu$M), lonidamine (10–10,000 $\mu$M) and 7ACC2 (5–500 $\mu$M). Data points represent the mean of three technical replicates of a typical experiment. The IC$_{50}$ measurements have also been independently replicated, three times for UK5099, Zaprinast and 7ACC2 (average IC$_{50}$: $9 \pm 7$, $18 \pm 8$ and $27 \pm 13$ $\mu$M, respectively) and two times for lonidamine (average IC$_{50}$: $118 \pm 24$ $\mu$M).

F   [$^{14}$C]-pyruvate homo-exchange inhibition by the TZDs, pioglitazone and rosiglitazone ($n = 6$).

Data information: Data have been independently replicated: (A) four biological repeats, (B) three biological repeats, (C, D) two biological repeats, (E, F) three biological repeats for UK5099, Zaprinast, 7ACC2 and two biological repeats for lonidamine and the TZDs. The error bars represent the standard error of the mean in (A) and the standard deviation in (C, D and F).

We were able to extend our analysis and compared the properties of the Mpc1/Mpc3 hetero-dimer with the Mpc3 homo-dimer, taking advantage of the CPM thermostability assay, where both proteins showed similar unfolding profiles (Fig 1D). It has been previously shown that binding events, creating new interactions between membrane proteins and inhibitors, result in an increased apparent melting temperature in thermostability analyses (Alexandrov *et al*, 2008; Crichton *et al*, 2015). If inhibitors bind to MPC, their interaction with the protein should also lead to a shift in thermostability upon binding. Indeed, at saturating concentrations, Zaprinast stabilised the Mpc1/Mpc3 hetero-complex by shifting the apparent melting temperature from 53.7 to 57°C (Fig 4C), consistent with its ability to inhibit transport. However, Zaprinast did not have an effect on the thermostability of the Mpc3 homo-dimer (Fig 4C), consistent with the inhibitor not interacting with Mpc3 alone. Unfortunately, we were not able to include UK5099 in this analysis as this

coloured compound quenches the fluorescent signal, complicating interpretation. Since Mpc1 did not show an unfolding curve in the CPM analysis (Fig 1D), we tested the effect of Zaprinast on Mpc1 via nanoDSF, using Mpc1/Mpc3 as a control (Fig 4D). While the Mpc1/Mpc3 hetero-complex was stabilised by Zaprinast, no stabilising shift was observed for Mpc1, but it is possible that the protein is not in a competent state. Overall, our results show that Mpc protomers can form homo-dimers, but they do not transport pyruvate and do not bind the inhibitor Zaprinast.

## Discussion

Despite the fundamental role of MPC in metabolism and disease, little was known about its composition and mechanism of transport and inhibition. We present here robust evidence that the

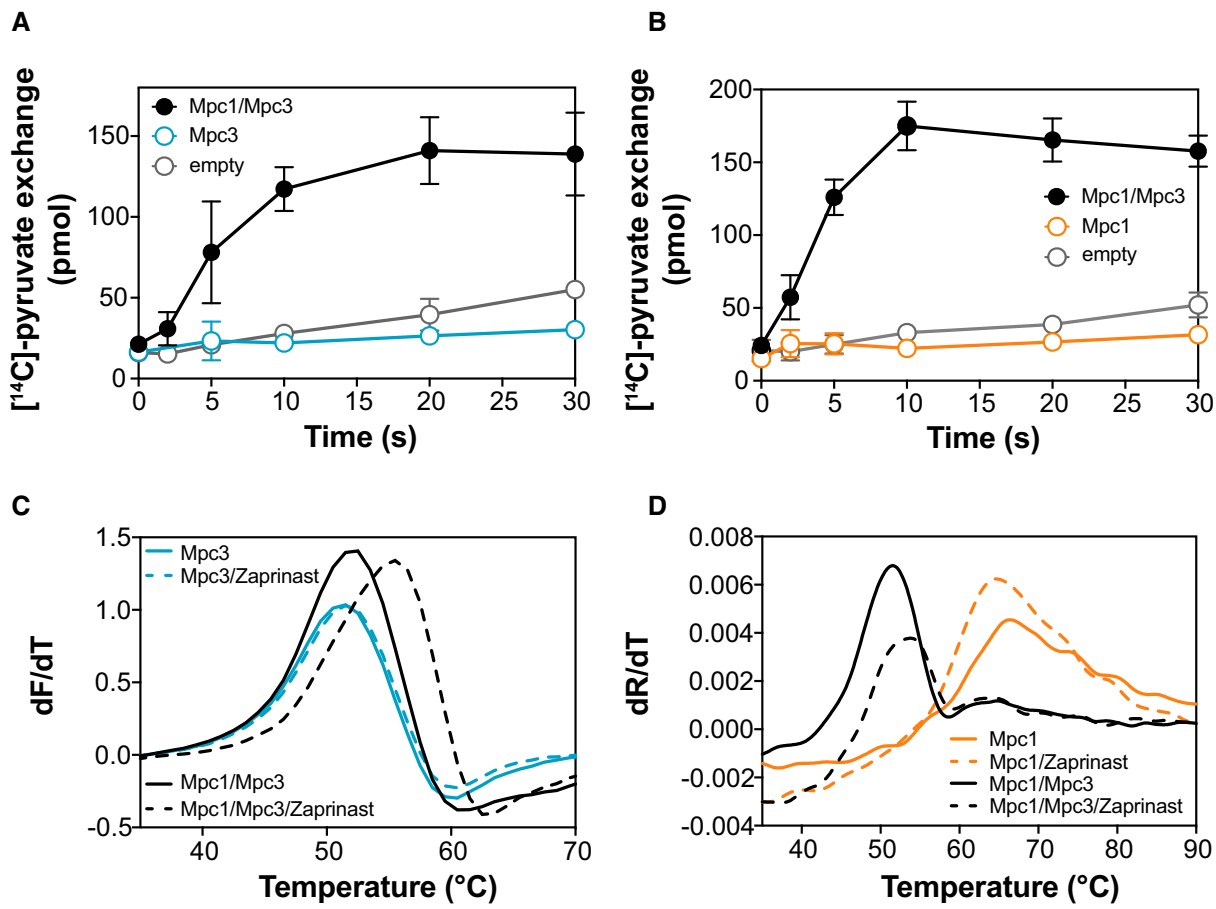

**Figure 4. Mpc1 or Mpc3 alone is not functional.**

A  Time course of pyruvate homo-exchange in proteoliposomes at a ΔpH of 1.6 was compared for Mpc3 (*n* = 4), the Mpc1/Mpc3 hetero-complex (*n* = 6) and empty liposomes (*n* = 4).

B  Time course of pyruvate homo-exchange at a ΔpH of 1.6 was compared for Mpc1 (*n* = 4), Mpc1/Mpc3 hetero-complex (*n* = 4) and empty liposomes (*n* = 4).

C  Thermostability analysis via cpm. In detergent solution, Zaprinast (250 μM) increased the thermostability of the Mpc1/Mpc3 hetero-complex (black lines; *n* = 3) but not of Mpc3 (blue lines; *n* = 3).

D  Thermostability analysis via nanoDSF. In detergent solution, Zaprinast (250 μM) increased the thermostability of the Mpc1/Mpc3 hetero-complex (black lines; *n* = 3) but not of Mpc1 (orange lines; *n* = 3).

Data information: Data have been independently replicated in two biological repeats. The error bars represent standard deviations.

mitochondrial pyruvate carrier is functional as a hetero-dimer. When the Mpc protomers from *S. cerevisiae* were expressed, purified and reconstituted into liposomes in combinations, the hetero-dimeric complexes were functional, whereas the individual proteins formed inactive homo-dimers.

Our results on the yeast Mpc are different than the published results on the human MPC proteins, where a co-expression and co-purification strategy did not lead to hetero-complex formation (Nagampalli *et al*, 2018). Specifically, in the presence of the co-expressed MPC1, the MPC2 protein was purified alone. This is not consistent with genetic and functional analyses of mammalian MPCs, including the human MPC, where co-expression of both MPC proteins was found to be necessary for mitochondrial pyruvate carrier activity (Bricker *et al*, 2012; Herzig *et al*, 2012; Compan *et al*, 2015; Vanderperre *et al*, 2016). We show here that the co-expressed yeast Mpc proteins co-purify to form folded and stable hetero-dimers in detergent solution. It is not possible to safely conclude on the oligomeric state of the mammalian MPCs, as it could differ between species, but our strategy can be applied to characterise the mammalian MPC complexes.

Establishing the oligomeric state of MPC is a first step to understand its bioenergetics and transport mechanism. Our results directly contradict the idea that the MPC proteins form multimeric complexes of 150 kDa, as proposed previously based on blue native gel electrophoretic analysis of the yeast complex Mpc1/Mpc2 (Bricker *et al*, 2012). We have shown here that within a large protein–detergent–lipid complex ($163.3 \pm 5.7$ kDa in our experimental setup), the protein contribution was only $30.8 \pm 1.3$ kDa, clearly corresponding to a hetero-dimer. It is likely that the anomalous migration of MPC on blue native gels is caused by binding of Coomassie stain to both the protein and the detergent–lipid micelle, as observed for other small membrane proteins (Crichton *et al*, 2013).

We also provide here strong evidence for the ability of the MPC hetero-dimers to transport pyruvate by measuring pyruvate homo-exchange activity, which was inhibited by previously proposed MPC inhibitors, such as UK5099, Zaprinast, lonidamine, 7ACC2 and the TZD rosiglitazone. Pyruvate exchange was dependent on the ΔpH, and the measured apparent affinity for pyruvate ($K_M$) was in line with other studies of pyruvate transport (Halestrap, 1975; Gray *et al*, 2016) and pyruvate homo-exchange in isolated mitochondria (Papa & Paradies, 1974; Paradies & Papa, 1975). These results demonstrate that the hetero-dimer is the functional unit of the yeast mitochondrial pyruvate carrier. Our experimental system provides a unique opportunity to study the detailed mechanism of pyruvate transport by the MPC hetero-complexes in yeast and creates new opportunities for the purification and reconstitution of the human MPC hetero-complex.

Consistent with previous observations that some MPC proteins homo-oligomerise (Bricker *et al*, 2012; Bender *et al*, 2015; Nagampalli *et al*, 2018), we found that Mpc1 and Mpc3 can be expressed and purified individually and we showed that Mpc3 forms stable homo-dimers, but only when it is expressed individually. Under normal conditions, it engages in hetero-complex formation with Mpc1. In our system, no transport activity could be detected for Mpc1 or Mpc3 homo-dimers in conditions where the hetero-dimer showed robust transport activity. The reason might be that pyruvate is an asymmetric substrate, requiring an asymmetric binding pocket for co-ordination, which cannot be provided by a homo-dimer.

Additionally, unlike the Mpc1/Mpc3 hetero-complex, the Mpc3 or Mpc1 alone did not interact with Zaprinast. Therefore, the yeast Mpc homo-dimers are not functional and they may not form a functional binding pocket.

Mpc1, Mpc2 and Mpc3 are highly homologous membrane proteins throughout the length of the conserved parts, strongly suggesting that the topology of all three must be very similar (Fig EV1). The observation that Mpc3 can form both a functional hetero-dimer with Mpc1 and a non-functional homo-dimer also indicates that these proteins must have similar structures. Investigation of the secondary structure and hydropathy profile indicates that the yeast Mpc proteins may have three transmembrane α-helices each (Fig EV1). The N-terminal region also has the propensity to form an α-helix, which is amphipathic in nature, whereas the secondary structure of the C-terminal region, which is highly variable in length, is unclear. The distantly related semi-SWEET transporters, which import sugars into bacteria (Xu *et al*, 2014; Lee *et al*, 2015), have a similar topology and also dimerise.

This work provides the basis for understanding the function of the MPC hetero-complex, the detailed mechanism of substrate transport and inhibition and the substrate specificity profile. Moreover, it will pave the way towards structural analysis of the complex. These studies will be important in light of the central role of MPC in metabolism, its involvement in diseases and its potential use as a drug target.

# Materials and Methods

### Molecular biology

The codon-optimised gene sequences for Mpc1 (UniProt: P53157), Mpc2 (UniProt: P38852) and Mpc3 (UniProt: P53311) from *S. cerevisiae* were synthesised (GenScript) and cloned into the bidirectional expression vector pBEVY-GU [gift from Charles Miller; Addgene plasmid # 51229 (Miller *et al*, 1998)]. For expression of *mpc1*, the cDNA was subcloned into the EcoRI/SacI sites and for *mpc2* or *mpc3* into the BamHI/XbaI sites. Where indicated, the sequences were designed to include a sequence coding for Factor Xa cleavage site (IEGR), followed by an octa-histidine tag at their C-termini. For expression of individual proteins, each single sequence was subcloned in the EcoRI/SacI sites of the same vector.

### Protein expression and mitochondrial preparations

The expression plasmids were transformed into an *mpc* triple deletion strain of *S. cerevisiae* (SHY15; Herzig *et al*, 2012) or into the W303-1B strain (MATα *leu2-3, 112 trp1-1 can1-100 ura3-1 ade2-1 his3-11,15*), using standard methods (Gietz & Schiestl, 2007). Successful transformants were selected on synthetic-complete uracil-dropout medium (Formedium) plates supplemented with 2% (w/v) glucose. Pre-cultures were grown in the same medium and used to inoculate 50 l of YPG medium containing 0.1% (w/v) glucose in an Applikon Pilot Plant 140-l bioreactor (Thangaratnarajah *et al*, 2014). Protein expression was induced for 3 h with 0.4% (w/v) galactose, after 20 h of growth in YPG. Mitochondrial isolation was performed, as previously described (Thangaratnarajah *et al*, 2014), using a DYNO-MILL (Willy A. Bachofen). Isolated

mitochondria were aliquoted, flash-frozen in liquid nitrogen and stored in −80°C until use.

## Affinity chromatography

All the protomers (Mpc1, Mpc2 and Mpc3) and the Mpc1/Mpc2 hetero-complex were expressed in and purified from mitochondria of the SHY15 strain. The Mpc1/Mpc3 hetero-complex was expressed in both SHY15 and W303-1B strains and yielded the same results with respect to the Mpc1/Mpc3 protein ratio and stability. The W303-1B strain was selected for large-scale purification of Mpc1/Mpc3 based on the total protein yield. Immediately prior to purification, 1 g of mitochondria was thawed and suspended in buffer containing 20 mM Tris–HCl, pH 7.4, 150 mM NaCl, 10% (v/v) glycerol, one Complete EDTA-free protease inhibitor cocktail tablet (Roche) and 1% (w/v) lauryl maltose neopentyl glycol (LMNG, Anatrace). Mitochondria were solubilised for 1.5 h at 4°C under gentle agitation and then clarified by ultracentrifugation at $205,000 \times g$ for 45 min. The supernatant was incubated for 2 h with nickel Sepharose beads (GE Healthcare), previously equilibrated with 20 mM Tris–HCl, pH 7.4, 150 mM NaCl, and then poured into an empty column (Bio-Rad). The column was initially washed with 20 column volumes of Buffer A [20 mM Tris–HCl, pH 7.4, 150 mM NaCl, 40 mM imidazole, 0.1% (w/v) LMNG, 0.1 mg/ml tetraoleoyl cardiolipin (TOCL)], followed by 20 column volumes of buffer B [20 mM Tris–HCl, pH 7.4, 150 mM NaCl, 5 mM $CaCl_2$, 0.1% (w/v) LMNG, 0.1 mg/ml TOCL]. Mpc1/Mpc3 was eluted from the column by on-column digestion for 12 h at 4°C with 10 μg of Factor Xa protease (New England Biolabs) per 1 g of mitochondria. Mpc1/Mpc2 was eluted after 1-h cleavage with Factor Xa with 10 μg of Factor Xa per 1 g of mitochondria. The mobile phase containing untagged MPC was separated from the resin with empty Proteus Midi spin columns (Generon) at $200 \times g$ for 5 min. Protein concentration was determined by the bicinchoninic acid assay (Thermo Fisher Scientific). Freshly purified protein was used for reconstitution into liposomes and for size-exclusion chromatography (SEC) coupled to multi-angle laser light scattering (MALLS), called SEC-MALLS.

## Size-exclusion chromatography

Analytical size-exclusion chromatography was performed on an ÄKTA Explorer (GE Healthcare) with a Superdex 200 10/300 GL column (GE Healthcare) equilibrated in SEC buffer [20 mM Tris–HCl, pH 7.4, 150 mM NaCl, 0.05% (w/v) LMNG, 0.05 mg/ml TOCL]. Nickel-purified proteins were injected without being concentrated onto the column at 0.3 ml/min, and 0.15 ml fractions were collected. The column was calibrated with a high molecular weight calibration kit (GE Healthcare) in the same buffer without detergent and lipid.

## Thermostability analysis using a thiol reactive probe

The assessment of protein stability was performed via thermal denaturation using a rotary quantitative PCR (qPCR) instrument as previously described (Crichton *et al*, 2015). In this method, cysteine residues, buried within the protein structure, become solvent exposed during denaturation in a temperature ramp and react with N-[4-(7-diethylamino-4-methyl-3-coumarinyl)phenyl]-maleimide (CPM) to form fluorescent-adducts. Briefly, a CPM working solution was

prepared by diluting the CPM stock (5 mg/ml in dimethyl sulfoxide) 50-fold into assay solution (20 mM Tris–HCl, pH 7.4, 150 mM NaCl, 5 mM $CaCl_2$, 0.1% (w/v) LMNG, 0.1 mg/ml TOCL) and incubated for 10 min at room temperature. For each analysis, carried out in triplicates, 3 μg of purified MPC was diluted into the same buffer to a final volume of 45 μl, to which 5 μl of the CPM working solution was added. When testing the effect of MPC inhibitors by thermostability shift assays, the protein was diluted in assay buffer containing the desired concentration of the inhibitor, and then, 5 μl of the CPM working solution was added. In each case, samples were incubated on ice for a further 10 min and then subjected to a temperature gradient from 25 to 90°C at 1°C increments with a 4-s hold, corresponding to a temperature ramp of 5.6°C/min. The fluorescence increase was monitored with the HRM channel of the machine (excitation at 440–480 nm and emission at 505–515 nm). Unfolding profiles were analysed with the Rotor-Gene Q software 2.3, and the peaks of their derivatives were used to determine the apparent melting temperature as a relative measure of protein stability.

## Thermostability analysis by nanoDSF

The protein stability was also assessed using dye-free nano differential scanning fluorimetry (nanoDSF), which monitors fluorescence changes due to altered environments of tryptophan and tyrosine residues during unfolding. Protein samples in buffer containing 20 mM Tris–HCl pH 7.4, 150 mM NaCl, 5 mM $CaCl_2$, 0.1% (w/v) LMNG, 0.1 mg/ml TOCL, in the presence or absence of the indicated concentrations of small-molecule inhibitors, were loaded into capillary tubes and subjected to a temperature gradient from 20 to 95°C with a temperature ramp of 5°C/min. The intrinsic fluorescence was measured using the NanoTemper Prometheus NT.48 instrument.

## Immunoblotting

Protein concentrations of crude mitochondrial preparations, isolated from the *S. cerevisiae* SHY15 (Herzig *et al*, 2012) or W303-1B strains, were determined by the bicinchoninic acid assay (Pierce). Twenty-five micrograms of mitochondria was subjected to SDS–PAGE (4–20% gradient gel), and proteins were electroblotted onto polyvinylidene difluoride (PVDF) membranes. Proteins were probed with polyclonal antibodies raised in hen against Mpc peptides (Agrisera) and detected with a rabbit anti-chicken (IgG) horseradish peroxidase conjugate (Sigma, A9046). The antibodies were raised against a synthetic peptide corresponding to residues 111–126 of Mpc1 and residues 40–54 of Mpc3. For detection of Mpc2, a mouse anti-his antibody (Roche, 04905318001) was used, followed by goat anti-mouse (IgG) horseradish peroxidase conjugate (Thermo Fisher Scientific, G21040). The signal was developed using the ECL reagent Western blot detection kit (GE Healthcare) and visualised on X-ray films.

## Mass determination by SEC-MALLS

SEC-MALLS analysis was performed with a Superdex 200 10/300 GL column (GE Healthcare) on an ÄKTA Explorer (GE Healthcare) coupled in line with a light scattering detector (Dawn HELEOS II, Wyatt Technologies) and a refractometer (Optilab T-rEX, Wyatt

Technologies). The Mpc1/Mpc3 complex or the individual Mpc3 protein was injected at 0.3 ml/min onto the Superdex 200 10/300 GL column equilibrated with 20 mM Tris–HCl, pH 7.4, 150 mM NaCl, 0.005% (w/v) LMNG and 0.005 mg/ml TOCL. All data were recorded and analysed with ASTRA 6.03 (Wyatt Technologies). Molecular weight calculations were performed using the protein-conjugate method (Slotboom *et al*, 2008) with the dn/dc value for protein of 0.185 ml/g and dn/dc value for LMNG-TOCL of 0.1675 ml/g (Thangaratnarajah *et al*, 2014). To determine the contribution of each protein to the overall protein–detergent–lipid complex, the extinction coefficients $\varepsilon_{A280}$ were calculated from the amino acid sequence using the ProtParam tool on the ExPaSy server (Gasteiger *et al*, 2005).

### Reconstitution in proteoliposomes

Egg L-α-phosphatidylcholine 99% (Avanti Polar Lipids) and tetraoleoyl cardiolipin (Avanti Polar Lipids) were mixed in a 20:1 (w/w) ratio, dried under a stream of nitrogen and washed once with methanol before being dried again. Lipids were hydrated in 20 mM Tris–HCl, pH 8.0 and 50 mM NaCl to a concentration of 12 mg/ml. Cold pyruvate to be internalised was added as a freshly made concentrated stock, where indicated. Lipids were solubilised with 1.2% (v/v) pentaethylene glycol monododecyl ether (Sigma), and freshly purified protein was added at a lipid-to-protein ratio of 250:1 (w/w) for Mpc1/Mpc3 and 125:1 for Mpc1/Mpc2. Samples were incubated on ice for 5 min, after which liposomes were formed by the step-wise removal of pentaethylene glycol monododecyl ether by five additions of 60 mg Bio-Beads SM-2 (Bio-Rad) with gentle mixing at 4°C at 20-min intervals. A final addition of 180 mg Bio-Beads was incubated with the samples overnight. Proteoliposomes were first separated from the Bio-Beads by passage through empty spin columns (Bio-Rad) and subsequently pelleted at $120,000 \times g$ for 60 min. The proteoliposomes were resuspended with a thin needle in 150 μl of their supernatant after the rest of it was removed.

### Pyruvate transport assays

The time course of pyruvate homo-exchange was measured at room temperature. The transport was initiated by diluting the proteoliposomes 200-fold into external buffer, containing 50 μM [$^{14}$C]-pyruvate (500,000 dpm, Perkin Elmer). The external buffers of different pH were (i) 20 mM MES, pH 5.4 and 50 mM NaCl; (ii) 20 mM MES, pH 6.4 and 50 mM NaCl; (iii) 20 mM Tris–HCl, pH 7.4 and 50 mM NaCl; and (iv) 20 mM Tris–HCl, pH 8.0 and 50 mM NaCl. The reaction (0–60 s) was terminated by rapid dilution into 8 volumes of ice-cold internal buffer (20 mM Tris–HCl, pH 8.0, 50 mM NaCl or 20 mM Tris–HCl, pH 7.4, 50 mM NaCl, as indicated), followed by rapid filtration through cellulose nitrate 0.45-μm filters (Millipore) and washing with an additional 8 volumes of buffer. The filters were dissolved in Ultima Gold scintillation liquid (Perkin Elmer), and the radioactivity was counted with a Perkin Elmer Tri-Carb 2800 RT liquid scintillation counter. The initial rate measurements to determine kinetic parameters were taken after 5 s of linear transport. Increasing concentrations of pyruvate were achieved by diluting the specific activity of [$^{14}$C]-pyruvate with unlabelled pyruvate. For inhibition of pyruvate transport various concentrations (as indicated in the Figure legends) of UK5099, Zaprinast, lonidamine, 7ACC2 or the indicated TZDs were added to the liposomes simultaneously with 50 μM radioactive substrate. The data analysis was performed with non-linear regression fittings using GraphPad Prism 7.0d [(Inhibitor) vs. response, variable slope]. The specific uptake rates were calculated based on the amount of protein used in reconstitutions, as determined by bicinchoninic acid assay. The biological repeats represent independent proteoliposome preparations using fresh protein from independent purifications.

### Peptide mass fingerprinting

SDS–polyacrylamide gel portions were subjected to in-gel proteolytic digestion with trypsin according to standard protocols. Extracted peptide mixtures were analysed with a 4800 MALDI-TOF/TOF mass spectrometer (Applied Biosystems). Peptide fragmentation spectra were matched to the NCBI non-redundant DNA sequence database version 20120611 (18480950 sequences; 6336030745 residues) using the software package Mascot version 2.4 (Matrix Science www.matrixscience.com). Peptide ion scores greater than 58 points indicate identity or extensive homology ($P < 0.05$).

## Data availability

Original data available on the Dryad Digital Repository. https://doi.org/10.5061/dryad.m173s71

**Expanded View** for this article is available online.

### Acknowledgements

We thank Drs. Ian Fearnley and Shujing Ding for total mass analysis performed at the MRC Mitochondrial Biology Unit and Dr. Chris Johnson for access to the NanoTemper Prometheus NT.48 at the MRC Laboratory of Molecular Biology. We also thank Dr. Shane Palmer for the large-scale fermentation and Dr. Martin S. King for providing valuable feedback on the manuscript. This work was supported by the Medical Research Council Grant MC_UU_00015/1 (to E.R.S.K.), the Swiss National Science Foundation 31003A_179421/1 (to J-C.M.) and the Oncosuisse grant KFS-4434-02-2018 (to J-C.M.).

### Author contributions

ST, CT and ERSK designed the research; ST, CT, VM and MEH performed the research; ST and CT analysed the data; ST, CT, J-CM and ERSK wrote the paper.

### Conflict of interest

The authors declare that they have no conflict of interest.

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
