## [Review Process File · The EMBO Journal]

The yeast mitochondrial pyruvate carrier is a hetero-dimer in its functional state

Sotiria Tavoulari, Chancievan Thangaratnarajah, Vasiliki Mavridou, Michael E. Harbour, Jean-Claude Martinou and Edmund R.S. Kunji

Review timeline:

Submission date:	25 th September 2018
Editorial Decision:	7 th November 2018
Revision received:	4 th February 2019
Editorial Decision:	7 th March 2019
Revision received:	13 th March 2019
Accepted:	20 th March 2019

Editor: Elisabetta Argenzio

Transaction Report:

1st Editorial Decision

7th November 2018

Thank you for submitting your manuscript on the characterization of mitochondrial pyruvate carrier in yeast to The EMBO Journal. Your study has been sent to three referees for evaluation, and we have now received reports from them, which are enclosed below for your information.

As you can see, the referees concur with us on the overall interest of your findings. However, they also raise some points that need to be addressed before they can support publication in The EMBO Journal. In particular, referees #1 and #2 request you to characterize pyruvate transport activity of the MPC1-MPC3 heterodimer. Also, referee #1 finds that differences between MCP protein oligomerization in yeast and mammals need to be discussed and points out that the specificity of the employed MPC inhibitor has to be assessed on the individual heterodimer subunits. Referee #3 asks you to test other MCP inhibitors by employing 14C-pyruvate exchange and thermostability assays.

Addressing these issues as suggested by the referees is required to warrant publication in The EMBO Journal. Given the overall interest of your study, I would like to invite you to revise the manuscript in response to the referee reports.

REFeree REPORTS

Referee #1:

Tavoulari et al have provided data on the characterisation of oligomers of the pyruvate carrier protein in the yeast *S. cerevisiae*. They have managed to purify hetero-oligomeric complexes from yeast and characterize them functionally in a reconstituted system. The question of the oligomeric state of the pyruvate carrier and how this is linked to its transport function is central to the understanding of the biological role of this protein. Since the discovery in 2012 of MPC proteins by two independent studies, several more studies appeared linking the MPC to cell metabolism in major diseases

(cancer, diabetes etc). However, a thorough analysis of the oligomeric state is missing. In the absence of a detailed 3D structure, the determination of the active oligomer state will be important for future studies. The work presented is generally well executed. The authors argue that the functional entity for MPC in *S cerevisiae* is a heterodimer between MPC1 and MPC3. The transport capacity of the heterodimer is sensitive to specific inhibitors, but the homodimer seems to be non-functional. The data are convincing, but there are a number of issues to be resolved. Also, the authors should tone down the statements that what they find in yeast is universally true for mammalian homologues.

1. Figure 2 shows that MPC1 and MPC3 form a hetero-dimeric complex using size exclusion chromatography. It is not known what residues or surfaces may contribute to the formation of a heterodimer, and hetero-dimer is a preferred state as opposed to a homodimer when the partner subunit is co-expressed. This should be addressed by some mutagenesis analysis.

This would be reinforced by mutagenesis in yeast to see (i) the effect of expressing one of the subunits in a mutated form incapable (or affected) in its hetero-dimer capacity and (ii) to see whether this translates also in affected transport of pyruvate *in vivo*.

2. It is an interesting result that the reconstituted heterodimer is sensitive to UK5099. However, the difference in the efficiency of inhibition between the mammalian and the yeast proteins reinforces the argument that the mammalian system may not necessarily oligomerize in the same manner as the one from yeast. The authors should discuss this possibility rather than dismissing the study of Nagampalli et al 2018 due to technical differences in the purification. To be fair, in the absence of an equivalent reconstituted system of the mammalian carriers one cannot draw safe conclusions about the properties of the mammalian carriers simply by analogy to the yeast system.

3. The use of the inhibitor Zaprinas is also interesting and adds to the functional analysis of the MPC. To see if the drug was specific for either the dimer interface or an individual subunit, a control using both monomeric proteins should be tested for thermal stability in the presence of the drug.

Minor point: In Figure 1B please indicate with arrows the MPC proteins and explain what the bands above the pure protein are.

Referee #2:

Mitochondrial pyruvate carrier (MPC) proteins facilitate the transport of pyruvate from the cytosol to the mitochondrial matrix. These are MPC1 and MPC2 in mammals and Mpc1, Mpc2 and Mpc3 in baker's yeast. The precise stoichiometry, oligomeric state and the functional unit of the MPC complexes have not been defined so far.

The work of Tavoulari et al. describes the successful co-expression, purification and characterization of the heterocomplexes Mpc1-Mpc2 and Mpc1-Mpc3 as well as the corresponding homodimers of Mpc1 and of Mpc3. The authors demonstrate that in principle MPC proteins from yeast are able to form both homo- and hetero-dimers. They also show that only the Mpc1-Mpc3 heterodimer (Mpc1-Mpc2 was not tested) is capable of transporting pyruvate and thus presents the functional unit of MPC.

Previous *in vivo* genetic study (e.g. Herzig et al., 2012) already indicated that MPCs function as hetero-multimeric complexes: Mpc1-Mpc2 under fermentative conditions and Mpc1-Mpc3 during respiration. Thus, the here presented *in vitro* biochemical approaches nicely confirm this hypothesis and extend our knowledge in the function of the MPC. However, I also think that this limits the novelty of the article in its present form. I however also think that these limitations could be overcome in a potential major revision.

Major comments:

1- Although the authors report on the successful Mpc1-Mpc2 heterodimer purification, they did not characterize its pyruvate transport activity. Especially in the light of the potential differential use of the Mpc1-Mpc2 and Mpc1-Mpc3 heterodimers during different metabolic conditions, an

experimental comparison of the transport properties of both complexes appears critical.

2- Homodimers do not exhibit pyruvate transport. How can this be explained? Is it possible to turn such a homodimer by directed mutagenesis into a transporter of pyruvate?

Referee #3:

In this Ms, the authors present a strong set of evidence supporting the heterodimeric nature of functional MPC. The experiments with an inducible bidirectional vector transduced in *mpc* deletion strain led to the purification of a stable Mpc1/Mpc3 1:1 heterocomplex. ¹⁴C-pyruvate measurements into proteoliposomes are convincing, and particularly relevant with differential pHi=8/pHe=7.2.

With their setup evaluating ¹⁴C-pyruvate exchange (figs 3E-F) and possibly thermostability (fig 4C) (if compound characteristics allow it), the authors should provide the scientific community working in the field with informations on the handful MPC inhibitors reported so far. Besides UK5099 and Zaprinast, thiazolidinediones (Proc Natl Acad Sci U S A. 2013 Apr 2; 110(14): 5422-5427.), lonidamine (<http://www.biochemj.org/content/473/7/929.long>) and aminocarboxycoumarins 7ACC2 (<https://www.nature.com/articles/s41467-018-03525-0>) have been reported. This information would considerably broaden the interest of the manuscript for the EMBO J readership.

1st Revision - authors' response

4th Februray 2019

Thank you for the opportunity to revise our manuscript in light of the reviewers' comments. We appreciate the positive evaluation and the helpful comments by all the reviewers. Following is a point-by-point response to each of the comments:

Referee #1

Tavoulari et al have provided data on the characterisation of oligomers of the pyruvate carrier protein in the yeast *S. cerevisiae*. They have managed to purify hetero-oligomeric complexes from yeast and characterize them functionally in a reconstituted system. The question of the oligomeric state of the pyruvate carrier and how this is linked to its transport function is central to the understanding of the biological role of this protein. Since the discovery in 2012 of MPC proteins by two independent studies, several more studies appeared linking the MPC to cell metabolism in major diseases (cancer, diabetes etc). However, a thorough analysis of the oligomeric state is missing. In the absence of a detailed 3D structure, the determination of the active oligomer state will be important for future studies. The work presented is generally well executed. The authors argue that the functional entity for MPC in *S. cerevisiae* is a heterodimer between MPC1 and MPC3. The transport capacity of the heterodimer is sensitive to specific inhibitors, but the homodimer seems to be non-functional. The data are convincing, but there are a number of issues to be resolved. Also, the authors should tone down the statements that what they find in yeast is universally true for mammalian homologues.

- 1) Figure 2 shows that MPC1 and MPC3 form a hetero-dimeric complex using size exclusion chromatography. It is not known what residues or surfaces may contribute to the formation of a heterodimer, and hetero-dimer is a preferred state as opposed to a homodimer when the partner subunit is co-expressed. This should be addressed by some mutagenesis analysis. This would be reinforced by mutagenesis in yeast to see (i) the effect of expressing one of the subunits in a mutated form incapable (or affected) in its hetero-dimer capacity and (ii) to see whether this translates also in affected transport of pyruvate in vivo.

Response: We agree with the reviewer that it would be interesting to elucidate the dimer interface. However, in the absence of structures to guide the mutagenesis analysis we would have to investigate a large number of mutant complexes (>100 to mutate one of two protomers), each requiring a technically challenging purification and functional analysis. The other problem is that this approach might lead to false positives and false negatives. Mutation of a single interface residue

might not be sufficient for disrupting the interface (a false negative) and thus a combination of different mutants might be required. If a mutation affects the structure, it will prevent dimerisation even when the residue is not in the interface (a false positive). Functional assays would not be conclusive, as the activity can be affected by other reasons, such as an impaired substrate binding and transport mechanism or by protein misfolding, none of which have anything to do with the dimerization interface. We hope that the reviewer will agree that obtaining structures of the heterodimers would be the preferred way to address this question, which is beyond the scope of this manuscript.

- 2) It is an interesting result that the reconstituted heterodimer is sensitive to UK5099. However, the difference in the efficiency of inhibition between the mammalian and the yeast proteins reinforces the argument that the mammalian system may not necessarily oligomerize in the same manner as the one from yeast. The authors should discuss this possibility rather than dismissing the study of Nagampalli et al 2018 due to technical differences in the purification. To be fair, in the absence of an equivalent reconstituted system of the mammalian carriers one cannot draw safe conclusions about the properties of the mammalian carriers simply by analogy to the yeast system.

Response: We agree with the reviewer that there might be differences between the yeast and the mammalian MPC complexes and that the same analysis needs to be performed on the mammalian complexes. However, we would like to note that previous studies on mammalian MPCs attributed functionality to the hetero-complexes and not to single proteins (Bricker *et al.*, 2012, Compan *et al.*, 2015, Herzig *et al.*, 2012, Vanderperre *et al.*, 2016). To comply with the comments of the reviewer, we have emphasised in the discussion that there are possible differences between the yeast and mammalian MPCs, including the oligomeric state, and we have removed our criticism on the human MPC purification.

- 3) The use of the inhibitor Zaprinst is also interesting and adds to the functional analysis of the MPC. To see if the drug was specific for either the dimer interface or an individual subunit, a control using both monomeric proteins should be tested for thermal stability in the presence of the drug.

Response: We thank the reviewer for bringing up this control. We had partially addressed it by testing the effect of Zaprinst on the thermostability of Mpc3 alone (Figure 4, panel C) and found that Mpc3 is not stabilized as the hetero-complex. In the CPM assay, the purified sample of Mpc1 does not show a denaturation profile (Figure 1), so cannot be used. However, we have now tested the effect of Zaprinst on Mpc1 with the nanoDSF (Figure 4, panel D) and found that there was no stabilizing shift on Mpc1 alone, further supporting that only the hetero-dimer can bind Zaprinst.

Minor point: In Figure 1B please indicate with arrows the MPC proteins and explain what the bands above the pure protein are.

Response: We apologise for not making clear that the samples in Figure 1B are not purified protein but crude mitochondrial preparations. We have now indicated that in the main text and the figure legend. We have also indicated the bands that are most likely Mpc proteins, based on molecular weight, with arrows. The smears detected above the Mpc bands are possibly due to non-specific binding of the polyclonal hen antibodies.

Referee #2:

Mitochondrial pyruvate carrier (MPC) proteins facilitate the transport of pyruvate from the cytosol to the mitochondrial matrix. These are MPC1 and MPC2 in mammals and Mpc1, Mpc2 and Mpc3 in baker's yeast. The precise stoichiometry, oligomeric state and the functional unit of the MPC complexes have not been defined so far.

The work of Tavoulari et al. describes the successful co-expression, purification and characterization of the heterocomplexes Mpc1-Mpc2 and Mpc1-Mpc3 as well as the corresponding homodimers of Mpc1 and of Mpc3. The authors demonstrate that in principle MPC proteins from yeast are able to form both homo- and hetero-dimers. They also show that only the Mpc1-Mpc3 heterodimer (Mpc1-Mpc2 was not tested) is capable of transporting pyruvate and thus presents the functional unit of MPC.

Previous *in vivo* genetic study (e.g. Herzig et al., 2012) already indicated that MPCs function as hetero-multimeric complexes: Mpc1-Mpc2 under fermentative conditions and Mpc1-Mpc3 during respiration. Thus, the here presented *in vitro* biochemical approaches nicely confirm this hypothesis and extend our knowledge in the function of the MPC. However, I also think that this limits the novelty of the article in its present form. I however also think that these limitations could be overcome in a potential major revision.

Response: We would like to thank the reviewer for the positive comments and for noting the misconception about MPCs being multimeric complexes. We think that our work is novel in terms of providing the precise stoichiometry, oligomeric state and functional unit of MPC.

Major comments:

1) Although the authors report on the successful Mpc1-Mpc2 heterodimer purification, they did not characterize its pyruvate transport activity. Especially in the light of the potential differential use of the Mpc1-Mpc2 and Mpc1-Mpc3 heterodimers during different metabolic conditions, an experimental comparison of the transport properties of both complexes appears critical.

Response: We agree that a comparison between the two alternative MPC complexes is important. In our original manuscript we have mentioned the problems with the stability of the Mpc1/Mpc2 purified protein. As we had anticipated, the reconstitution of Mpc1/Mpc2 into liposomes was challenging mainly due to the low yield of the purified protein and its stability. However, by introducing modifications in the purification and reconstitution protocols we managed to measure pyruvate transport by the Mpc1/Mpc2 hetero-complex, shown now in Figure EV4, panel D. The low yield, concentration and stability of the purified Mpc1/Mpc2 (one order of magnitude less than Mpc1/Mpc3) was a limiting factor for the reconstitution procedures. Given the incredible difficulty in handling of this complex, it is not possible to carry out an extensive comparison.

2) Homodimers do not exhibit pyruvate transport. How can this be explained? Is it possible to turn such a homodimer by directed mutagenesis into a transporter of pyruvate?

Response: The reviewer raises an interesting question. Since pyruvate is an asymmetric substrate it is expected to bind to an asymmetric binding site, which can be provided by a hetero-dimer but not by a homo-dimer. Other transporters (i.e. some ABC transporters), are also known to function as obligate hetero-dimers. Since the binding site, dimerization interface, and conformational changes have not been defined for MPC to guide mutagenesis, it would be nearly impossible to experimentally convert the homo-dimers into active pyruvate transporters at this point.

Referee #3:

In this Ms, the authors present a strong set of evidence supporting the heterodimeric nature of functional MPC. The experiments with an inducible bidirectional vector transduced in *mpc* deletion strain led to the purification of a stable Mpc1/Mpc3 1 : 1 heterocomplex. ¹⁴C-pyruvate measurements into proteoliposomes are convincing, and particularly relevant with differential pHi=8/pHe=7.2.

With their setup evaluating ¹⁴C-pyruvate exchange (figs 3E-F) and possibly thermostability (fig 4C) (if compound characteristics allow it), the authors should provide the scientific community working in the field with informations on the handful MPC inhibitors reported so far. Besides UK5099 and Zaprinast, thiazolidinediones (Proc Natl Acad Sci U S A. 2013 Apr 2; 110(14): 5422-5427), lonidamine (<http://www.biochemj.org/content/473/7/929.long>) and aminocarboxycoumarins 7ACC2 (<https://www.nature.com/articles/s41467-018-03525-0>) have been reported. This information would considerably broaden the interest of the manuscript for the EMBO J readership.

Response: We would like to thank the reviewer for the positive comments on our work. We have tested the suggested compounds for their ability to inhibit pyruvate transport by the yeast Mpc1/Mpc3 reconstituted in proteoliposomes. In addition to UK5099 and Zaprinast, we have now measured IC₅₀ values for lonidamine and 7ACC2 and we have incorporated IC₅₀ data for all compounds in Figure 3, panel E. We have also tested two different thiazolidinediones, pioglitazone and rosiglitazone, for their ability to inhibit pyruvate exchange. Even in high concentrations of 500

μM pioglitazone had no effect on transport activity. At the same high concentration, however, rosiglitazone inhibited pyruvate transport to 52%. We could not test higher concentrations because beyond the 500 μM these compounds were not soluble in our buffer system. As we have emphasized in the discussion section, the inhibitor co-ordination appears different between the yeast and mammalian MPC complexes and we cannot draw any conclusions for the ability of the TZDs to inhibit the human MPC.

References

- Bricker DK, Taylor EB, Schell JC, Orsak T, Boutron A, Chen YC, Cox JE, Cardon CM, Van Vranken JG, Dephoure N, Redin C, Boudina S, Gygi SP, Brivet M, Thummel CS, Rutter J (2012) A mitochondrial pyruvate carrier required for pyruvate uptake in yeast, Drosophila, and humans. *Science* 337: 96-100
- Herzig S, Raemy E, Montessuit S, Veuthey JL, Zamboni N, Westermann B, Kunji ER, Martinou JC (2012) Identification and functional expression of the mitochondrial pyruvate carrier. *Science* 337: 93-96
- Compan V, Pierredon S, Vanderperre B, Krznar P, Marchiq I, Zamboni N, Pouyssegur J, Martinou JC (2015) Monitoring Mitochondrial Pyruvate Carrier Activity in Real Time Using a BRET-Based Biosensor: Investigation of the Warburg Effect. *Mol Cell* 59: 491-501
- Vanderperre B, Cermakova K, Escoffier J, Kaba M, Bender T, Nef S, Martinou JC (2016) MPC1-like Is a Placental Mammal-specific Mitochondrial Pyruvate Carrier Subunit Expressed in Postmeiotic Male Germ Cells. *J Biol Chem* 291: 16448-16461

2nd Editorial Decision

7th March 2019

Thank you for submitting a revised version of your manuscript. It has now been seen by one of the original referees, whose comments are appended below.

As you will see, s/he finds that all criticisms have been sufficiently addressed and recommends the study for publication. However, before we can officially accept the manuscript, I kindly ask you to review and approve the text edits to the legends made by our production/data editors (in attachment to this e-mail).

REFEREE REPORTS

Referee #2:

The authors have addressed my comments in writing and experimentally. I appreciate the difficulties of the purification of membrane proteins and that the authors undertook the effort to nevertheless try to characterize the Mpc1-Mpc2 heterodimer.

Corresponding Author Name: Sotiria Tavoulari

Manuscript Number: EMBOJ-2018-100785